# Satellite observations of unprecedented phytoplankton blooms in the Maud Rise Polynya, Southern Ocean

Babula Jena, and Anilkumar Narayana Pillai

ESSO - National Centre for Polar and Ocean Research, Ministry of Earth Science, Government of India, Vasco-da-Gama, India.

*Correspondence to*: Babula Jena (bjena@ncpor.res.in)

**Abstract.** Appearance of phytoplankton blooms with in the sea-ice cover has large importance considering the upper ocean primary production that controls the biological pump with the implications for atmospheric $CO_2$ and global climate. Satellite derived chlorophyll-*a* concentration showed the unprecedented phytoplankton blooms in the Maud Rise polynya, Southern Ocean with chlorophyll-*a* reached up to 4.67 mg m$^{-3}$. Multi-satellite data indicated that the bloom appeared for the first time in the entire mission records started since 1978. Argo float located in the polynya edge provided evidence of bloom condition in austral spring 2017 (chlorophyll-*a* up to 5.47 mg m$^{-3}$) compared to the preceding years of prevailed low chlorophyll-*a*. The occurrence of bloom was associated with the supply of nutrients into the upper ocean through the Ekman upwelling (driven by wind stress curl and cyclonic ocean eddies), and improved light condition up to 61.9 Einstein m$^{-2}$ day$^{-1}$. The net primary production from the Aqua-Moderate Resolution Imaging Spectroradiometer chlorophyll-based algorithm showed that the Maud Rise polynya was as productive as the Antarctic coastal polynyas with the carbon fixation rates reached up to 415.08 mg C m$^{-2}$ day$^{-1}$. The study demonstrates how the phytoplankton in the Southern Ocean (specifically over the shallow bathymetric region) would likely respond in the future under a warming climate condition and continued melting of Antarctic sea-ice since 2016.

## 1 Introduction

Antarctica sea-ice has moderately increased during the satellite era from 1979 to 2015, with the regional heterogeneity that comprises of both increasing and decreasing pattern in different sectors (Turner et al., 2017). However, anomalously record lowest sea-ice extent and area observed since three successive years from 2016 to 2018 with the maximum melting occurred in 2017 (Parkinson, 2019). Amid the pronounced melting, the largest and most prolonged Maud Rise (MR) open ocean polynya since the 1970s reappeared on 14 September 2017 (~$9.3 \times 10^3$ km$^2$) that expanded maximum on 1 December 2017 (~$298.1 \times 10^3$ km$^2$) and existed for 79 days (Jena et al., 2019). Appearance of the polynyas plays an important role for the oceanic phytoplankton and primary production that controls the biological pump of the ocean (Arrigo and Dijken, 2003; Shadwick et al., 2017), apart from its importance for marine mammals and birds (Labrousse et al., 2018; Stirling, 1997), global heat-salt fluxes (Tamura et al., 2008), Antarctic bottom water properties (Zanowski et al., 2015), and atmospheric

circulation (Weijer et al., 2017). However, due to their spatial dimension, the polynyas are generally not represented well in the large-scale climate models, limiting the capability of simulating and projecting polynya related biophysical changes under future climate change scenario (Li et al., 2016).

The Southern Ocean (SO) is known as the largest high-nutrient low-chlorophyll (HNLC) area of the global ocean. Since the past 50 years, the loss of ice shelves and glaciers retreat around the Antarctic Peninsula has increased at least 24,000 km$^2$ in surface area of new open water that was rapidly colonised by new phytoplankton blooms, with new benthic and marine zooplankton communities in the SO (Peck et al., 2010). In the background of HNLC, the occurrence of polynyas can enhance the chlorophyll-*a* (chl-*a*) concentration (a proxy for phytoplankton biomass) due to the increase in surface area of new open waters and growth season of the phytoplankton (Kahru et al., 2016). The bloom occurrence in the SO has been linked with the oceanographic features such as jet streams, meanders and mesoscale eddies, which can lead to increased iron and silicate supply by the ocean upwelling (Strass et al., 2002), thereby improving co-limitation of nutrient and light for phytoplankton growth (Hoppe et al., 2017) . Oceanic eddies have been found to regulate chl-*a* variability in the SO with higher (lower) values observed for the cyclonic (anticyclones) eddies (Kahru et al., 2007). The polynyas of the Amundsen and Ross Seas have high primary productivity that contribute to the SO carbon dioxide ($CO_2$) sink (Alderkamp et al., 2012; Arrigo et al., 2008a; Arrigo and Alderkamp, 2012; Yager et al., 2012). The primary productivity of these regions reaching up to 3 g C m$^{-2}$ d$^{-1}$, roughly 10 folds more than the SO mean productivity (Arrigo and Dijken, 2003). The high productivity values in the polynya have been attributed to the supply of iron from the upwelling of iron rich deep water (Planquette et al., 2013), sediment diffusion or resuspension followed by upwelling (Ardelan et al., 2010), atmospheric inputs (Cassar et al., 2007; Wagener et al., 2008), melting of sea-ice (Lannuzel et al., 2010; van der Merwe et al., 2011), iceberg delivered glacial debris (Raiswell et al., 2008), and melting of ice-shelves (Pritchard et al., 2009; Wåhlin et al., 2010). The Amundsen polynya is one of the productive polynyas of the Antarctica with the satellite derived chl-*a* (2.2 mg m$^{-3}$) are 40% greater than the Ross Sea Polynya (1.5 mg m$^{-3}$)(Schofield et al., 2015). Although the polynyas are believed as the sites of phytoplankton blooms in spring (Arrigo and Dijken, 2003) and acts as sinks of atmospheric $CO_2$ because of both physical-chemical processes and biological activity (Bates et al., 1998; Mu et al., 2014), very little is known about the MR polynya due to its rare appearance. In this paper, we report first evidence of occurrence of phytoplankton bloom in the MR polynya from satellite derived ocean color data and the Argo float. Further, the role of physical processes for the occurrence of bloom in the polynya is examined using relevant physical oceanographic data, followed by its likely implication for ocean-atmospheric exchange of $CO_2$.

## 2 Materials and Methods

In order to understand the impact of bathymetry on the phytoplankton biomass, the MR seamount was mapped using bathymetric raster data ($21601 \times 10801$ pixels) from Earth Topography One Arc-Minute Global Relief Model, 2009 (ETOPO1) (www.ngdc.noaa.gov). The raster data were converted to polyline features with a contour interval of 500 m for

showing the extent of the seamount (Fig. 1a). Level-3 monthly composite of satellite derived near-surface chl-*a* imageries were used from the Nimbus-7 Coastal Zone Colour Scanner (CZCS), Sea-viewing Wide Field-of-view Sensor (SeaWiFS),

Aqua-Moderate Resolution Imaging Spectroradiometer (Aqua-MODIS), and Visible Infrared Imaging Radiometer Suite (VIIRS), as per the availability of data from 1978 to 2017 (Fig. 1b-e). Level-2 Aqua-MODIS ascending passes were processed (relatively cloud free data) to generate the high spatial resolution (~1 km) chl-*a* images during 25 October (14:45 hours UTC), 06 November (15:05 hours UTC) and 21 November 2017 (14:25 hours UTC) (Fig. 2). We used a standard chl-*a* retrieving algorithm that uses combination of both lower and higher range of chl-*a* retrieval as described in the Algorithm

Theoretical Basis Document (ATBD) from the NASA Earth Observing System Project Science Office (https://oceancolor.gsfc.nasa.gov/atbd/chlor_a/). In this study, we have used the criteria of chl-*a* $\geq$ 0.8 mg m$^{-3}$ (Fitch and Moore, 2007), for defining a phytoplankton bloom after considering the underestimation tendency of chl-*a* measurement from satellite observations over the Southern Ocean (Jena, 2017).

In order to analyze the Aqua-MODIS derived net primary production (NPP), we have validated three ocean-color based models such as the vertically generalized production model (VGPM), *Eppley*-VGPM, and carbon-based productivity model (CbPM) for selecting the best model for the study region. We evaluated the performance of these models by comparing with the in-situ NPP estimated using $^{13}$C tracer during the Indian scientific expedition to the Southern Ocean in 2009. The locations of in-situ NPP observations during the austral summer (February to April 2009) are presented in figure 3a. The in-

situ NPP from 11 observations range from about 85.04 to 923.83 mg C m$^{-2}$ day$^{-1}$. The detail method of $^{13}$C measurement was documented in the previous work (Gandhi et al., 2012). The VGPM was developed for estimation of NPP from chlorophyll accounting into temperature dependency of chlorophyll-specific photosynthetic efficiency (Behrenfeld and Falkowski, 1997b). The *Eppley*-VGPM makes use of an exponential function developed from changes in growth rates of phytoplankton over varied temperature ranges for a wide variety of species (Eppley, Richard, 1972). Further, a new CbPM model was

developed that uses the backscattering coefficients and chlorophyll-to-carbon ratios for estimation of phytoplankton carbon biomass and phytoplankton growth rates, respectively (Westberry et al., 2008). The model based NPP values were available in weekly time scale with a spatial resolution of ~4 km. The pixel values from the models were extracted around each in-situ observations of NPP to generate the matchups for the validation strategy, a method adopted by several authors (Jena, 2017; Johnson et al., 2013). The comparative statistical analysis suggested that the scatters were much better in the case of *Eppley*-

VGPM estimated NPP (Fig. 3c) than those in the case of VGPM (Fig. 3b) and CbPM (Fig. 3d). A bias of -26.21 mg C m$^{-2}$ day$^{-1}$ for *Eppley*-VGPM obtained NPP value was much better than those obtained from the VGPM (bias = 104.40 mg C m$^{-2}$ day$^{-1}$) and CbPM (bias = 94.14 mg C m$^{-2}$ day$^{-1}$) (Table 1). The NPP values from VGPM and CbPM indicated significant overestimations. The coefficient of correlation (*r*) and standard error (SE) for *Eppley*-VGPM NPP values (*r* = 0.82 and SE = 116.16 mg C m$^{-2}$ day$^{-1}$) were better than that obtained from the VGPM (*r* = 0.82 and SE = 203.69 mg C m$^{-2}$ day$^{-1}$) and

CbPM (*r* = 0.66 and SE = 142.84 mg C m$^{-2}$ day$^{-1}$). Results suggested the *Eppley*-VGPM based NPP values match reasonably

well with the in-situ NPP. Therefore, we used the *Eppley*-VGPM model for the present study taking Aqua-MODIS as the input.

**Table 1.** Validation of ocean-colour based models (VGPM, *Eppley*-VGPM, and CbPM) with in-situ net primary production (mg C m$^{-2}$ day$^{-1}$) estimated using $^{13}$C tracer during the scientific expeditions to the Southern Ocean in 2009. CbPM- carbon based productivity model, VGPM- vertically generalized production model.

| | *r* (coefficient of correlation) | Standard error | Bias | *p*-value |
|---|---|---|---|---|
| VGPM | 0.82 | 203.69 | 104.40 | 0.001 |
| Eppley-VGPM | 0.82 | 116.16 | -26.21 | 0.001 |
| CbPM | 0.66 | 142.84 | 94.14 | 0.026 |

We used monthly sea-ice concentration (SIC) data (September to November 2017) from the Special Sensor Microwave Imager Sounder (SSMIS) with spatial resolution of 25 km acquired from the National Snow and Ice Data Center (NSIDC) (Data id-G02135, Version 3). The data were generated using the NASA Team algorithm, which converts satellite derived brightness temperatures to gridded SIC (Cavalieri, D. J., C. L. Parkinson, P. Gloersen, 1997). A detail description about the sensor characteristics, sea-ice processing methods, synoptic coverage, resolution, projection, and validation of sea-ice retrieval from passive microwave sensors are given in earlier work (Fetterer et al., 2016). The polynya was interpreted when the pixel values found to be less than or equal to 15% of SIC (Fig. 4a-c) (Jena et al., 2019). In order to examine the role of oceanic processes for the formation of the phytoplankton bloom in the polynya, we used relevant physical oceanographic data. Metop-Advanced Scatterometer (ASCAT) wind stress curl and Ekman upwelling data (Pond, S., 1983) were acquired from the National Oceanic and Atmospheric Administration (NOAA) Coast watch (https://coastwatch.pfeg.noaa.gov) at a spatial resolution of 0.25° x 0.25° (Fig. 4g-l). Oceanic eddies were identified from the sea surface height anomaly (SSHA) and geostrophic currents (0.25° x 0.25°) derived from multi-mission merged satellite altimeter data (https://las.aviso.altimetry.fr/) (Fig. 4d-f) (Jena et al., 2019). Although the dipole structure of cyclonic and anticyclonic eddies was observed in the MR polynya, cyclonic eddies dominated the flow pattern in the region during the event. Therefore, we focused on the cyclonic eddies because they can upwell the deep warm and nutrient rich water to the upper ocean for the chl-*a* enhancement. The optimal interpolated sea surface temperature (OI SST) data (9 × 9 km) obtained from Remote Sensing Systems (www.remss.com), which was produced after merging of the microwave (cloud penetration capabilities) and infrared SST (high spatial resolution) using an OI scheme (Reynolds and Smith, 1994) (Fig. 4m-o). In order to understand the vertical structures of biophysical parameters, we used an Argo float (ID-5904468) data that had remained in the MR polynya from 2015 to 2017 (http://www.argo.ucsd.edu/) (Fig. 1a). The Argo based partial pressure of $CO_2$ (p$CO_2$) in the water column were calculated from a Deep-sea DuraFET pH sensor after using an existing algorithm for total alkalinity (Johnson et al., 2016). The uncertainty in the derived value is about 11 µatm at p$CO_2$ of 400 µatm (~2.7%), considering the combined contribution from the pH sensor, the alkalinity estimate, and carbonate system equilibrium

constants (Williams et al., 2017). The monthly incident shortwave radiation was acquired from the European Center for Medium-Range Weather Forecast (ECMWF) (grid resolution of 0.25°) during January 1979-December 2017. Monthly anomalies of shortwave radiation for September-November 2017 was computed relative to a 38-year climatology (1979-2016).

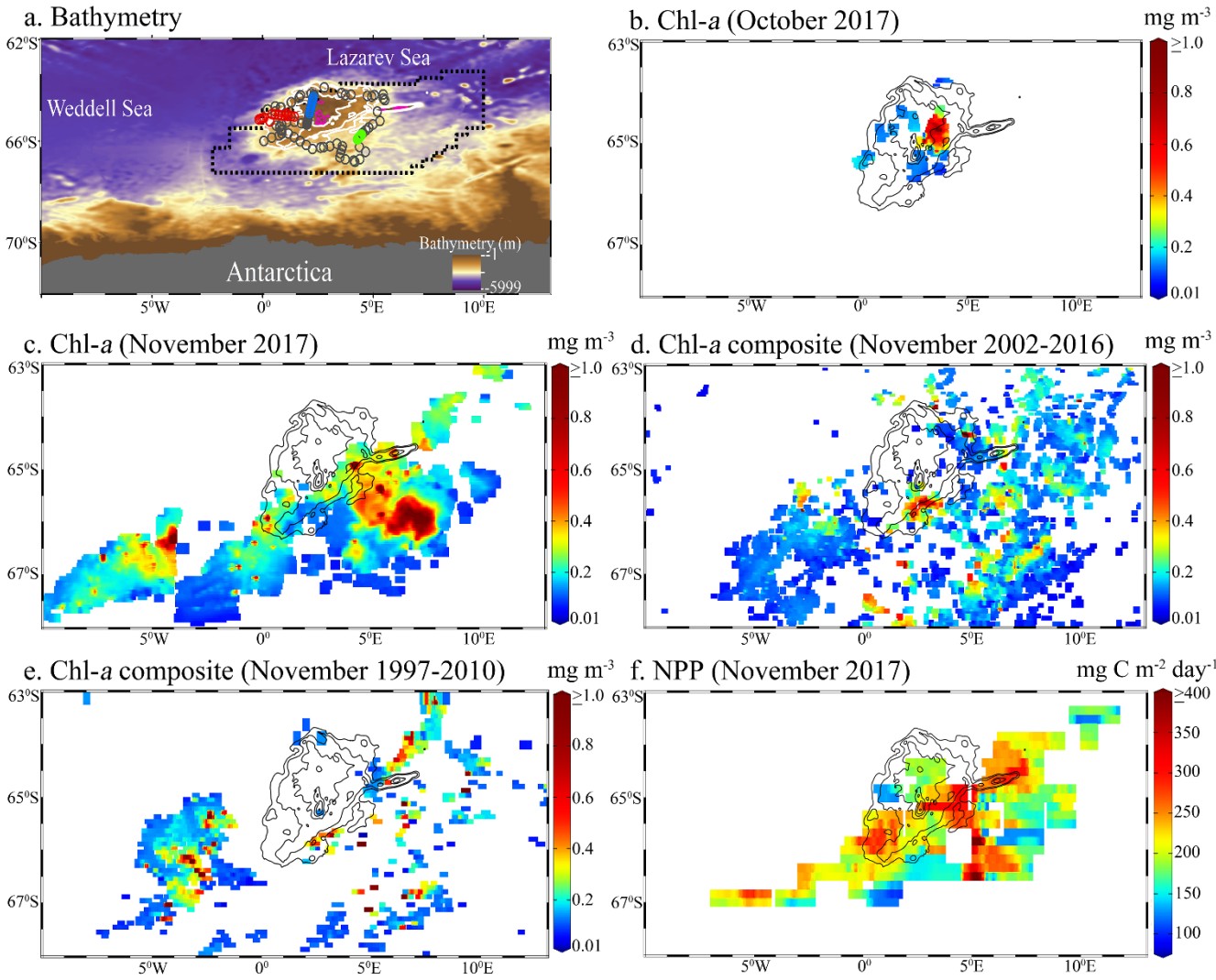

**Figure 1**. (a) Bathymetry map of the Maud Rise from Earth Topography One Arc-Minute Global Relief Model, 2009. Pink lines show the depth contours shallower than 2000 m and other white contours are spaced by 500 m with deeper values. Dashed polygon shows the extent of the polynya during 21 November 2017. Circles represents the location of an ARGO float (ID-5904468) from 19 January 2015 to 18 March 2018. Red, green and blue circles show the float location from August to December, respectively for 2017, 2016 and 2015. (b) Monthly mean chlorophyll-*a* (chl-*a*) from Aqua-Moderate

Resolution Imaging Spectroradiometer (Aqua-MODIS) during October 2017. (c) Monthly mean chl-*a* from Aqua-MODIS during November 2017. (d) Long-term composite of Aqua-MODIS chl-*a* (2002-2016) for November. (e) Long-term composite of SeaWiFS chl-*a* (1997-2010) for November. (f) Monthly mean daily net primary productivity (NPP) computed from the *Eppley*-vertically generalized production model for November 2017. The polyline features in figures b to f shows the extent of the Maud Rise seamount with a contour interval of 500 m.

## 3 Results and discussion

### 3.1 Phytoplankton bloom within the polynya

Although a large polynya was formed within the MR sea-ice cover during September 2017, no phytoplankton bloom observed in the satellite record. The polynya extent was nearly static from September to October and accompanied with a small patch of bloom (chl-*a* up to 3.48 mg m$^{-3}$) centred at 3.77°E and 64.72°S (Fig. 1b; 4b), which remains otherwise covered by the sea-ice. Prior to the October 2017 event, no chl-*a* was observed for the month of October from 1978 to 2016 even after considering entire data records of CZCS, SeaWiFS, Aqua-MODIS, and VIIRS. During November 2017, the polynya was enlarged and shifted southeastward with the high chl-*a* concentration reached up to 4.66 mg m$^{-3}$ (Fig. 1c; 4c). The bloom was formed approximately between 4°E to 8°E, and 64.5°S to 66.5°S. Prior to the November 2017 event, the satellite derived chl-*a* observation was scarce (SeaWiFS and MODIS) and missing (CZCS and VIIRS) for the month of November from 1978 to 2016. Figures 1d and 1e shows the climatological composite of chl-*a* observations in November, respectively for Aqua-MODIS (2002-2016) and SeaWiFS (1997-2010). The scarce and missing observations were mainly due to the presence of seasonal sea-ice cover and cloud cover on the MR. The result suggests that the observed bloom from October to November 2017 had appeared for the first time within the MR polynya in the records of satellite observations since 1978 (Fig. 1b-c). Even though the monthly composite images shown the evidence of blooms, we processed Level-2 high spatial resolution scenes of Aqua-MODIS that provided more information on this unprecedented phytoplankton bloom. Several selected scenes that has relatively better coverage showed a patch of bloom on 25 October, followed by a wide band of bloom during 06 November and 21 November 2017 (Fig. 2). The chl-*a* values reached as high as 4.67 mg m$^{-3}$ on 6 November 2017 (Fig. 2b). High diffuse attenuation coefficient (Kd 490) observed up to 0.39 m$^{-1}$ and 0.37 m$^{-1}$ during October and November, respectively, which is an indicator of sediment resuspension and bloom condition in the MR polynya (Table 2). The previously reported highest chl-*a* concentration in the Antarctic Polynya have been identified in the Amundsen Sea (coastal polynya) with the values reached about 6.98 mg m$^{-3}$ (Arrigo and Dijken, 2003). The bloom in the MR polynya was also tracked by a robotic Argo float (ID-5904468) that had remained at the north-western edge of the polynya (Fig. 1a; 5a). Result shows enhanced chl-*a* values from September to November 2017. The bloom condition was initiated on 25 October 2017 with the chl-*a* maxima up to 1.27 mg m$^{-3}$ (36 m depth) at 0.86°E and 64.98°S (Fig. 5a). The chl-*a* value reached up to 1.31 mg m$^{-3}$ (41 m depth) and 1.73 mg m$^{-3}$ (36 m depth), respectively on 4 November and 14 November 2017. Further on 24

November 2017, the values reached as high as 5.47 mg m$^{-3}$ (11 m depth) at 1.43°E and 65.04°S. In order to check whether this observed bloom is a seasonal or an episodic feature of the MR, we analyzed the Argo float data during two preceding

years of 2015 and 2016 when the sea-ice was covered. Analysis shows that the bloom was absent and the chl-*a* value found to be rather low during October and November for 2015 and 2016 (Fig. 5b,c). Thus, the result confirms that the observed bloom in 2017 was an unprecedented feature considering both the Argo float and multi-sensor satellite data.

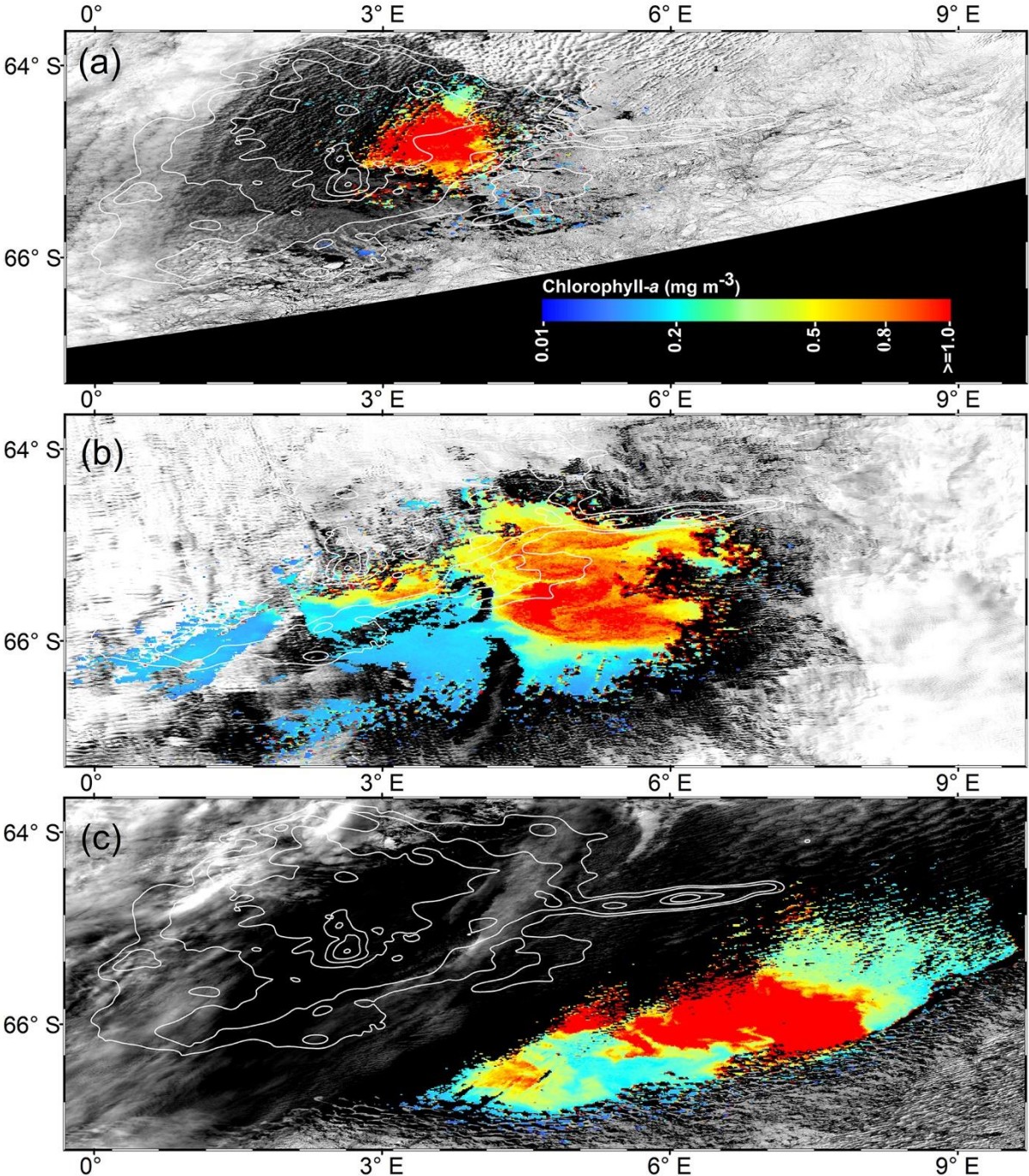

**Figure 2**. High spatial resolution (~1 km) Aqua-MODIS ascending passes during (a) 25 October (14:45 hours), (b) 06 November (15:05 hours), and (c) 21 November 2017 (14:25 hours), showing the unprecedented phytoplankton blooms in the Maud Rise polynya. The white contours show the extent of the Maud Rise seamount.

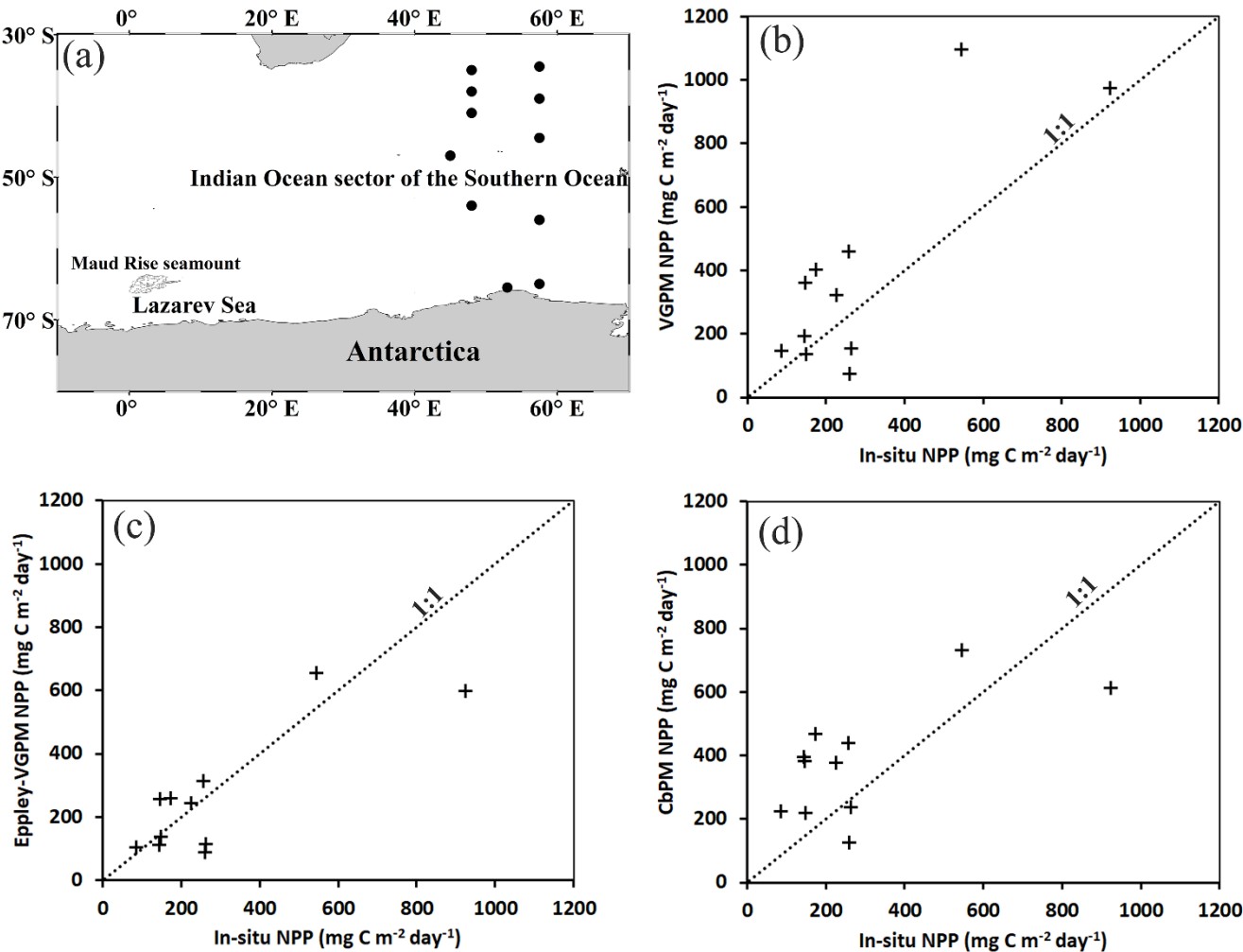

**Figure 3**. (a) Circles showing the locations of in-situ net primary production (NPP) from [13]C tracer during the Indian scientific expedition to the Southern Ocean (February to April 2009). Scatter plots between in-situ NPP and (b) VGPM, (c) *Eppley*-VGPM, (d) CbPM NPP estimations. NPP- net primary production, CbPM- carbon based productivity model, VGPM- vertically generalized production model.

## 3.2 Causes of the observed bloom formation

Generally, the phytoplankton biomass remains low in the SO which is mainly ascribed to the lack of micronutrient iron apart from strong zooplankton grazing pressure, light and silicate limitation (de Baar and Boyd, 1999; Boyd et al., 2001; Gall et al., 2001; Selph et al., 2001). The input of iron-enriched atmospheric dust from the continents to the SO is the lowest in the

world's oceans (Duce and Tindale, 1991); and the oceanic sources of iron from the deep water and vertical diffusion of iron
through the water column have been reported as the likely pathways of iron supply to the upper ocean (Jena, 2016; Tagliabue et al., 2014). The occurrence of phytoplankton bloom is possible over the shallow regions of MR seamount where the doming of isotherm/isopycnal can bring deeper high-nutrient water above the seamount where it may be utilised with a conducive environment of light availability and water column stability (White and Mohn, 2002). Analysis of bathymetric data indicated the peak of the MR seamount is located at 2.63°E, 65.23°S, that rises from the abyssal plain of ~5200 m to the
shallowest depth of ~968 m (Fig. 1a), influencing the local upliftment of thermocline and nutrient enriched deep-water (Jena et al., 2019; Mashayek et al., 2017; Muench et al., 2001; Roden, 2013). However, the oceanic processes that can bring subsurface nutrients to the sea surface have an important role for the formation of phytoplankton bloom. In order to examine the role of oceanic processes, we used satellite derived physical oceanographic data as shown in Fig. 4.

Analysis of monthly SSHA and corresponding geostrophic currents showed the presence of a large cyclonic eddy with a diameter of ~220 km in the vicinity of MR seamount during September 2017 (Fig. 4d). The center of the cyclonic eddy was located approximately at 3.84°E and 64.47°S, closely matching with the center of the polynya having warm SST (-1.35°C) compared to its peripheral cold SST of -1.79°C (Fig. 4d; 4m). The polynya extent was nearly static from September to October. During November, the polynya expanded southeastward in conjunction with the movement of cyclonic eddy
accompanied by a pool of warm SST and the phytoplankton bloom (Fig. 1c; 4f; 4o). The eddy was located approximately at 3.96°E and 66.5°S in November. Even though a dipole structure of cyclonic and anticyclonic eddies was observed in the polynya, a large cyclonic eddy dominated the flow pattern. The location of cyclonic eddy matching well with the annular halo of warm SST and patch of phytoplankton bloom in the polynya (Fig. 1b-c; 4d-f; 4m-o). In addition, we find that the polynya surface was associated with persistent negative wind stress curl (Fig. 4g-i) that induced upwelling of sub-surface
water to the sea surface during September-November 2017 (Fig. 4j-l). Generally, the water column on the MR seamount is characterized by the presence of a cold fresh layer in the upper ocean separated from a lower warm saline layer by a weak pycnocline (Jena et al., 2019; de Steur et al., 2007). The combined influence of the cyclonic eddy and negative wind stress curl brings up the warm thermocline water into the sea surface through Ekman upwelling that results in a pool of warm SST at the polynya center (Fig. 4m-o). Depth-Latitude cross section of the Copernicus Marine Environment Monitoring Service
(CMEMS) global analysis and forecast data on potential temperature data at a polynya location (along 4.7°E) provided evidence that the subsurface warm water was ventilated and brought closer to the upper ocean from the thermocline (upward doming of isotherms) during September through November 2017 (Jena et al., 2019). The Argo float located at the edge of the polynya also provided evidence on the upliftment of thermocline during September 2017 (Fig. 6a). Ocean upwelling is known to supply dissolved iron to the upper ocean (Klunder et al., 2014; Rosso et al., 2014), preferably at the shallow
bathymetry of less than 1 km at the MR seamount (Graham et al., 2015). Synchronously, along with the availability of light in October and November, the observed mechanism triggered a bloom condition in the MR polynya (Fig. 1b-c; 2a-c). ARGO float indicated mixed layer warming on the Maud Rise during spring 2016 and 2017 (Fig. 6). The upwelling of high saline

and warm water into the mixed layer facilitated the sea-ice melting. The melting of sea-ice leads to the development of shallow mixed layer due to the accumulation of freshwater in the upper ocean. Therefore, we observed lower values of salinity in the mixed layer with increased stability of the water column (Fig. 6). The development of shallow mixed layer improved the light availability in the upper ocean and the condition was favourable for the growth of phytoplankton. Even though the Ekman upwelling was evident in September, the bloom did not appear in the polynya region under low light condition up to 12.6 Einstein $m^{-2}$ $day^{-1}$. However, the bloom was appeared in October-November 2017 under the influence of Ekman upwelling and improved light condition up to 36.1 and 61.9 Einstein $m^{-2}$ $day^{-1}$, respectively for October and November (Table 2). Analysis of incident shortwave radiation data shows record highest gain of values in the polynya region during September-November 2017, considering the 38-year time series starting from 1979 through 2016 (Fig. 7). The observed anomalous gain in net shortwave radiation is possibly due to the early loss of sea ice cover.

Computation of NPP using the *Eppley*-VGPM model indicated the carbon fixation rates in the MR polynya varied between 60.08 and 374.07 mg C $m^{-2}$ $day^{-1}$, with an average value of 169.51 mg C $m^{-2}$ $day^{-1}$ for October 2017 (Table 2). The NPP increased in November that ranged from 101.43 to 415.08 mg C $m^{-2}$ $day^{-1}$, averaging 208.44 mg C $m^{-2}$ $day^{-1}$ with the highest rate being observed at 5.16°E and 66.58°S. The observed values in the polynya remained within the previously reported range for the Polar Frontal Zone of the SO (100-6000 mg C $m^{-2}$ $day^{-1}$) (Hoppe et al., 2017; Korb and Whitehouse, 2004; Mitchell and Holm-Hansen, 1991; Moore and Abbott, 2000; Park et al., 2010). The results from Aqua-MODIS observations in the Antarctic coastal polynyas indicated that the NPP values ranged from 34.3 to 911.9 mg C $m^{-2}$ $day^{-1}$ during November 2017 with the highest rate being observed at Sulzberger Bay polynya (Ross Sea) at 155.33°W and 76.08°S (Table 3). The NPP values in the MR polynya remained within the range of similar values observed for the coastal polynyas. The NPP values varied from 90 to 760 mg C $m^{-2}$ $day^{-1}$ for 37 coastal polynyas around the Antarctica (Arrigo and Dijken, 2003). Even though the phytoplankton bloom was appeared in the MR polynya with the NPP values similar to those of coastal polynyas, the spatial variation of NPP did not follow always the same pattern of chl-*a* (Figs. 1c; 1f). The observed pattern has been attributed to the effect of phytoplankton pigment composition and packaging (Bricaud et al., 2004; Ciotti et al., 2002; Jena, 2017; Lohrenz et al., 2003; Marra et al., 2007; Morel and Bricaud, 1981). The primary production in the upper ocean is a function of chl-*a*, availability of light, nutrients, phytoplankton-specific absorption coefficient (capacity of light absorption), and the efficiency of phytoplankton to convert the absorbed light for the carbon fixation (Behrenfeld and Falkowski, 1997a). However, the capacity of light absorption and the quantum yield of photosynthetic carbon fixation would vary from one phytoplankton community to another (Claustre et al., 2005). Although, the primary production in the Antarctic coastal polynyas are known to be dominated by prymnesiophytes (Phaeocystis antarctica) or diatoms (Arrigo et al., 2008b), the data on the phytoplankton community structure and their spectral characteristics are not available for the analysis in order to quantify the rate of carbon fixation for individual community.

**Table 2**. Net primary production and bio-optical parameters during the occurrence of Maud Rise polynya in October and November 2017. Values for November 2017 are given within brackets. NPP: Net primary production, Chl-*a*: Chlorophyll-*a*, Eu: Euphotic depth, PAR: Photosynthetically available radiation, Kd: Diffuse attenuation coefficient for downwelling irradiance, SST: Sea surface temperature.

| | Minimum | Maximum | Mean | Standard deviation |
|---|---|---|---|---|
| NPP (mg C $m^{-2}$ $day^{-1}$) | 60.08 (101.43) | 374.07 (415.08) | 169.51 (208.44) | 84.04 (50.90) |
| Chl-*a* (mg $m^{-3}$) | 0.07 (0.06) | 3.48 (4.67) | 0.29 (0.28) | 0.26 (0.20) |
| Eu (m) | 27.12 (8.35) | 84.24 (109.56) | 53.72 (56.90) | 13.59 (12.49) |
| PAR (Einstein $m^{-2}$ $day^{-1}$) | 6.27 (13.80) | 36.10 (61.90) | 17.79 (31.43) | 6.86 (8.21) |
| Kd 490 ($m^{-1}$) | 0.03 (0.02) | 0.39 (0.37) | 0.06 (0.06) | 0.03 (0.02) |
| SST (°C) | -1.80 (-1.80) | -1.25 (-1.31) | -1.67 (-1.65) | 0.12 (0.14) |


**Table 3.** Net primary production (mg C $m^{-2}$ $day^{-1}$) for some coastal polynyas around the Antarctica in November 2017. The values in the parentheses indicates locational information.

| | Minimum | Maximum | Mean | Standard deviation |
|---|---|---|---|---|
| Amundsen Bay, Enderby Land | 34.3 (50.25°E,66.75°S) | 55.9 (50.58°E,67.08°S) | 44.7 | 6.5 |
| Barrier, Prydz Bay | 161.8 (79.25°E,67.08°S) | 505.5 (80.25°E,67.08°S) | 308.5 | 93.2 |
| Vincennes Bay | 53.17 (108.83°E,66.83°S) | 68.9 (108.66°E,66.91°S) | 61.5 | 5.3 |
| Wrigley Gulf, Amundsen Sea | 52.5 (125.66°W,73.33°S) | 74.0 (125.58°W,73.41°S) | 67.2 | 6.8 |
| Sulzberger Bay, Ross Sea | 251.9 (154.33°W,75.91°S) | 911.9 (155.33°W,76.08°S) | 606.6 | 143.8 |

Further, Argo data were utilized to find the linkage between the observed bloom and the ocean $pCO_2$ condition. Analysis of
Argo data indicated low $pCO_2$ values that reached as low as 372.8 µatm (Fig. 5d) corresponding to the occurrence of bloom during October-November 2017 (Fig. 5a). The $pCO_2$ values declined during the occurrence of bloom in comparison with the period of non-bloom condition in August-September 2017, 2015 and 2016 (Fig.5). The coefficient of correlation (*r*) between the $pCO_2$ and chl-*a* was -0.56 ($p < 0.01$) during August-September 2017 (Fig. 8a). The relationship improved ($r = -0.82$, $p < 0.01$) and the spatial pattern closely matched together during the bloom condition in October-November 2017 (Fig. 8b; 5a-d).
The best relationship observed between the $pCO_2$ and chl-*a* when the data was log transformed ($r = -0.94$, $p < 0.01$). The observed low $pCO_2$ values in the polynya was likely due to the presence of chl-*a* bloom with high NPP, which has potential to drive $CO_2$ fluxes from the atmosphere to the ocean after forming a pressure gradient. This biological pumping process in

the polynya could play an important role for lowering the atmospheric $CO_2$ through transferring of atmospheric $CO_2$ to the ocean and subsequently into the ocean sediments. However, it is important to mention that the air-sea exchange of $CO_2$ is driven by the $pCO_2$ gradient, solubility of $CO_2$ in the seawater (function of ocean temperature and salinity), and gas transfer velocity (function of wind speed and SST) (Williams et al., 2017). Follow-up research works are required in the future to quantify the contribution from physical and biological processes for explaining the air-sea exchange of $CO_2$ in the MR polynya and its likely role in regulating the global climate (Gordon and Comiso, 1988; Li et al., 2016).

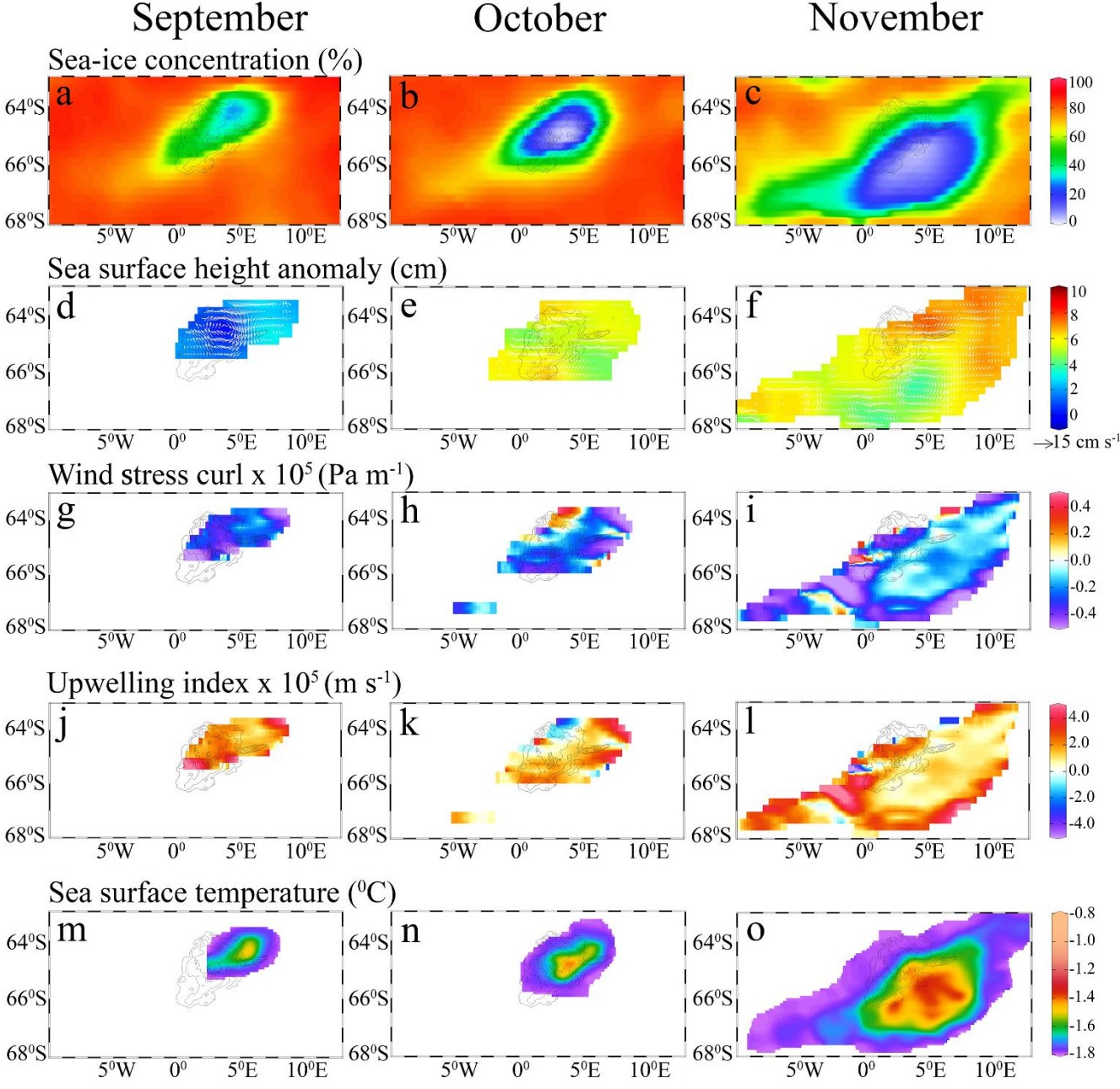


**Figure 4**. Monthly maps show (a-c) sea-ice concentration, (d-f) sea surface height anomaly and geostrophic current velocity (white arrows), (g-i) wind stress curl, (j-l) upwelling index, and (m-o) sea surface temperature variability during the appearance of polynya from September to November 2017.

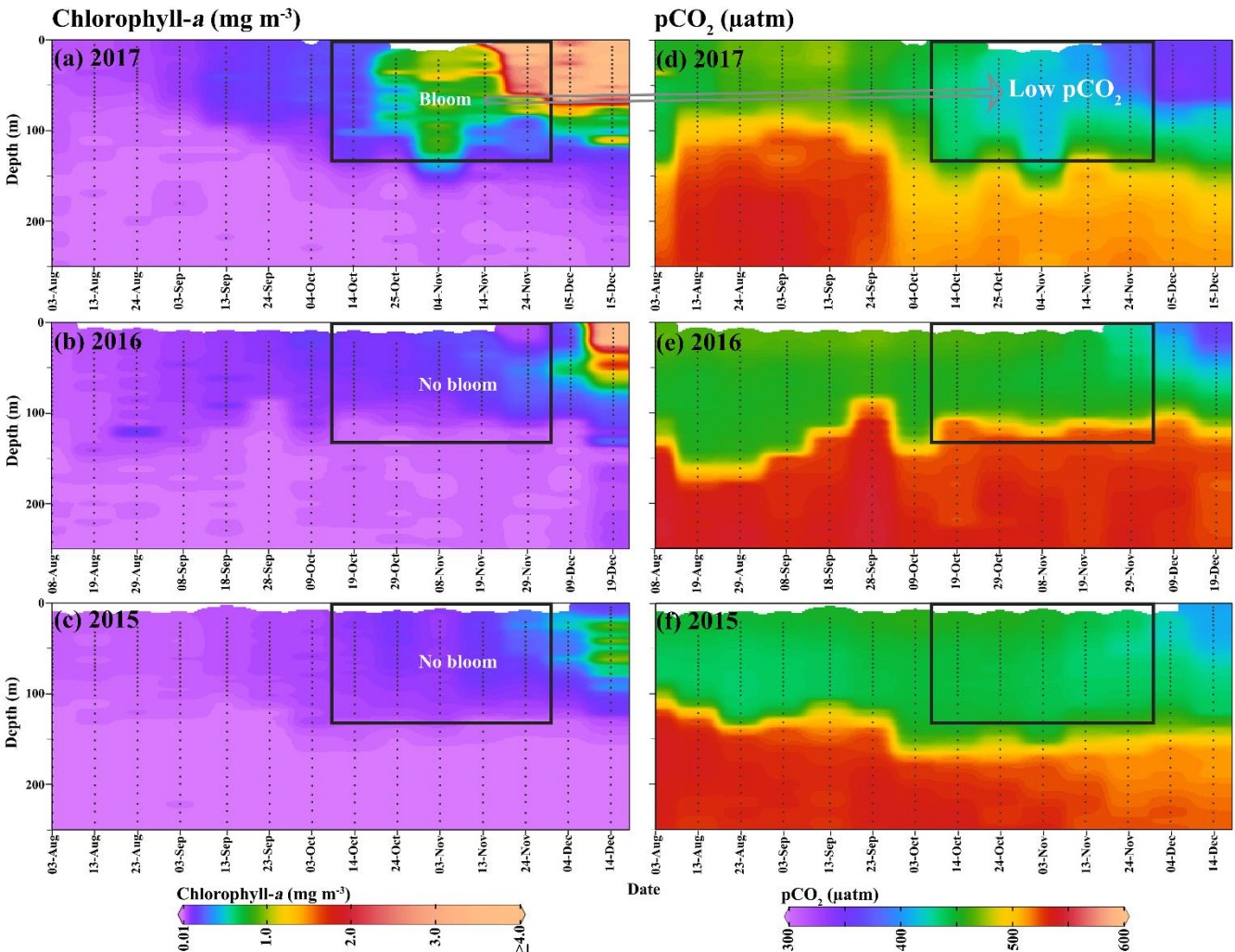

**Figure 5**. An Argo float (ID-5904468) located on the Maud Rise seamount shows profiles of (a-c) chlorophyll-*a*, and (d-f) pCO$_2$ from August to December (2015-2017). Marked rectangle in figure-a shows the bloom condition from October to November 2017, and the bloom was absent during respective period of two preceding years (2015 and 2016). Low pCO$_2$ values observed corresponding to the bloom occurrence.

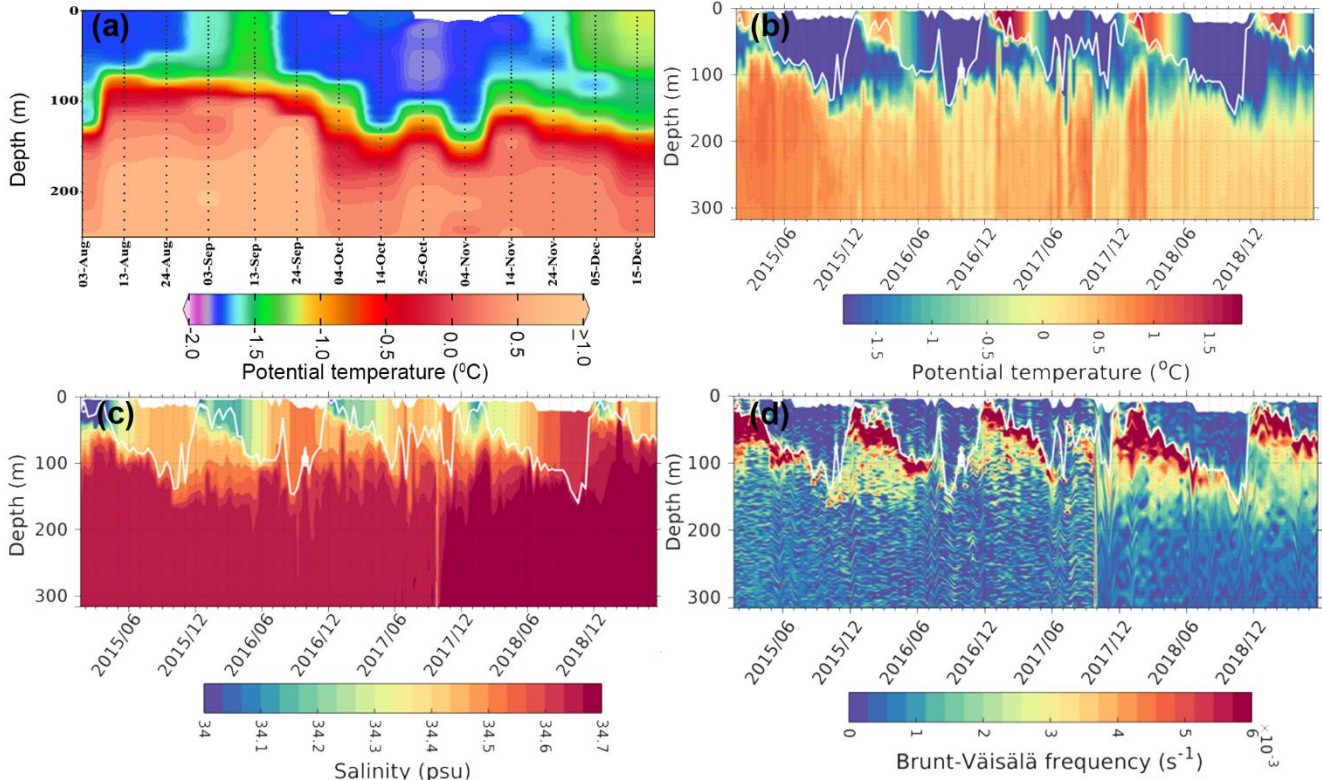

**Figure 6**. Profiles from an Argo float-ID-5904468 located at the edge of the Maud Rise polynya shows (a) potential temperature during August-December 2017, (b) potential temperature during 2015-2019, (c) salinity, and (d) static stability. White solid line shows the variability of mixed layer depth. The mixed layer was computed as the uppermost level of uniform potential density ($\sigma\theta$) at the depth where the density in the upper level varies by 0.01 kg m$^{-3}$ with reference to the surface (Kaufman et al., 2014).

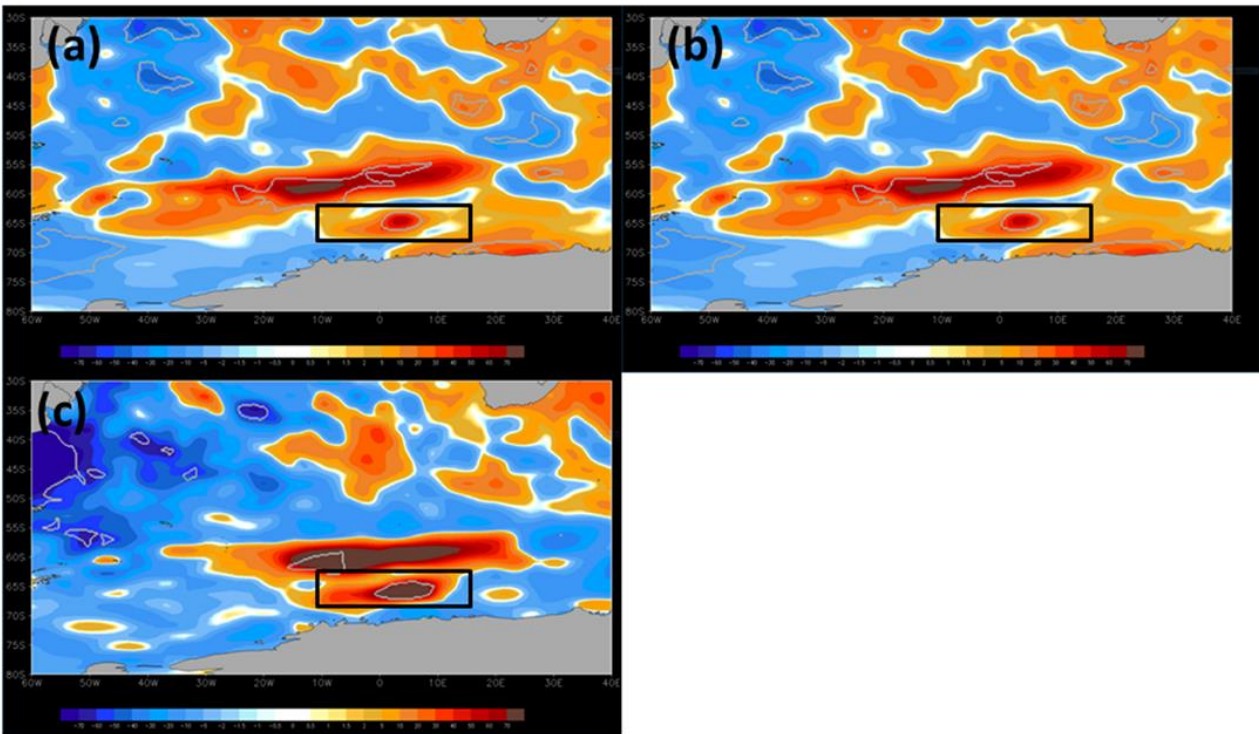

**Figure 7**. Monthly anomalies of incident shortwave radiation (W m$^{-2}$) for (a) September, (b) October, and (c) November 2017 in the Maud Rise polynya (black rectangles). The anomalies were computed relative to a 38-year climatology (1979-2016). The regions within grey polylines shows the record level shortwave radiation in 2017 that lies outside of values from 1979 to 2016.

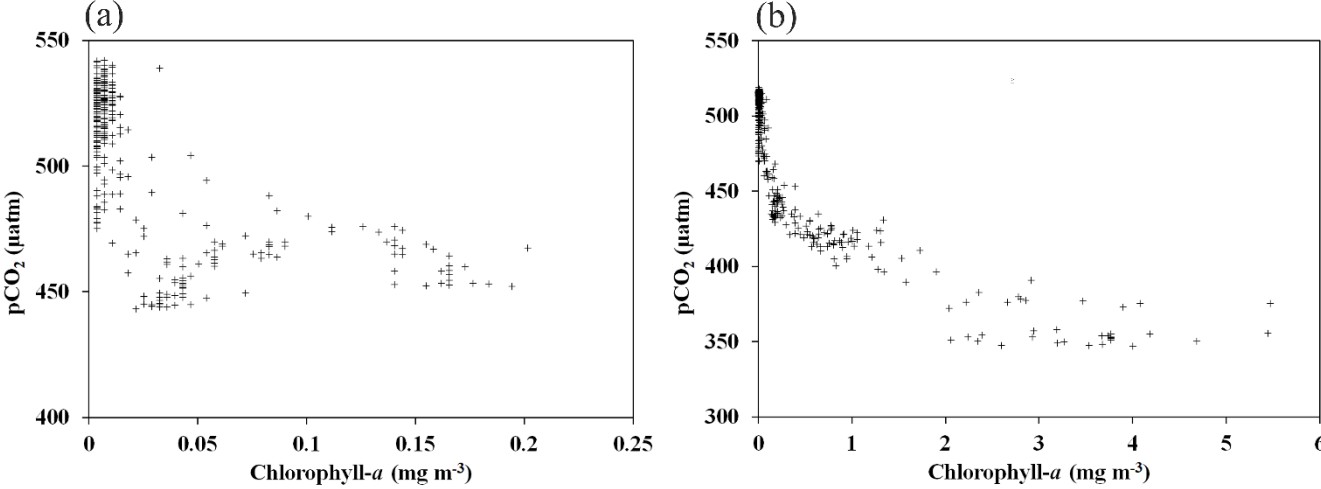

**Figure 8**. Argo data were utilized to find the relationship between the chlorophyll-*a* and the oceanic $pCO_2$ condition. (a) The coefficient of correlation (*r*) between the $pCO_2$ and chl-*a* was found to be -0.56 ($p < 0.01$) during August-September 2017. (b) The relationship improved ($r = -0.82$, $p < 0.01$) during the bloom condition in October-November 2017.

## 4 Summary and conclusion

In this article, we have shown that the phytoplankton bloom occurred on the MR seamount during the appearance of polynya in spring 2017. Analysis of multi-sensor satellite data from CZCS, SeaWiFS, MODIS, and VIIRS indicated that the bloom appeared for the first time in the satellite records since 1978. Since there is no previous report of its occurrence in the MR polynya, we have examined additional data from Argo float for a firm evidence. The ARGO float located at the north-western edge of the polynya provided evidence of bloom condition from October to November 2017 compared to the

preceding years of 2015 and 2016 when the sea-ice was covered at the surface with low chl-*a*. We find that the combined influence of seamount and physical processes are accountable for the formation of the observed bloom. The presence of a seamount on the MR leads to upliftment of thermocline and nutrient enriched deep water that could fertilize the upper ocean through support of upwelling process. During the austral winter and spring 2017, the supply of nutrient to the upper ocean arises through Ekman upwelling driven by a large cyclonic ocean eddy and the persistent negative wind stress curl. Even

though the Ekman upwelling was evident in September 2017, the bloom did not appear in the polynya due to prevailing low irradiance as expected in an austral winter. However, the bloom was appeared in austral spring (October-November 2017) under the influence of Ekman upwelling and improved light condition that favored for the phytoplankton photosynthesis and growth. Low $pCO_2$ condition prevailed in the polynya due to the presence of chl-*a* bloom with high NPP that can lead to sinking of atmospheric $CO_2$ fluxes into the ocean. The observed phytoplankton bloom reported in this article has large

importance considering the HNLC status of the SO.

Studies have shown intensification of polar cyclone activities due to the poleward shifting of the extratropical cyclone track in the background of a warming climate condition (Francis et al., 2019; Fyfe, 2003). As the polar cyclones are known to trigger the occurrence of polynyas (Francis et al., 2019; Jena et al., 2019; Turner et al., 2017) (through advection of moist-

warm air from extratropic, and sea-ice divergence), the frequency of polynya event is likely to be increased in the future (including over the MR) under a warming climate condition. The likelihood for the occurrence of the polynya is quite high with a background of anomalous upper ocean warming and sea-ice loss, similar to the events that occurred in the Antarctic sea-ice from 2016 to 2019 (Fig. 9). Indeed, the Weddell Sea and MR polynya has reappeared in 23 November 2018 that lasted till 12 December 2018 as observed from SSMIS (Fig. 10). With the frequent reoccurrence of polynya on the MR, the

associated physical processes could possibly modify the region into a productive environment and likely to have impact on the regional ecosystem and carbon cycle. The occurrence of polynya and phytoplankton blooms in the MR may lead it to a site of potential sink of atmospheric $CO_2$ through biological pumping and can be a major source of carbon and energy for the

regional food web. The spatial dimension of the bloom in a polynya might be small; however, it is necessary to monitor and understand as many important features of the Antarctic marine ecosystem in order to understand its complete role in the global biogeochemical cycle. The study demonstrates how the phytoplankton in the Southern Ocean (specifically over the shallow bathymetric region) would likely respond in the future under a warming climate condition and continued melting of Antarctic sea-ice since austral spring 2016.

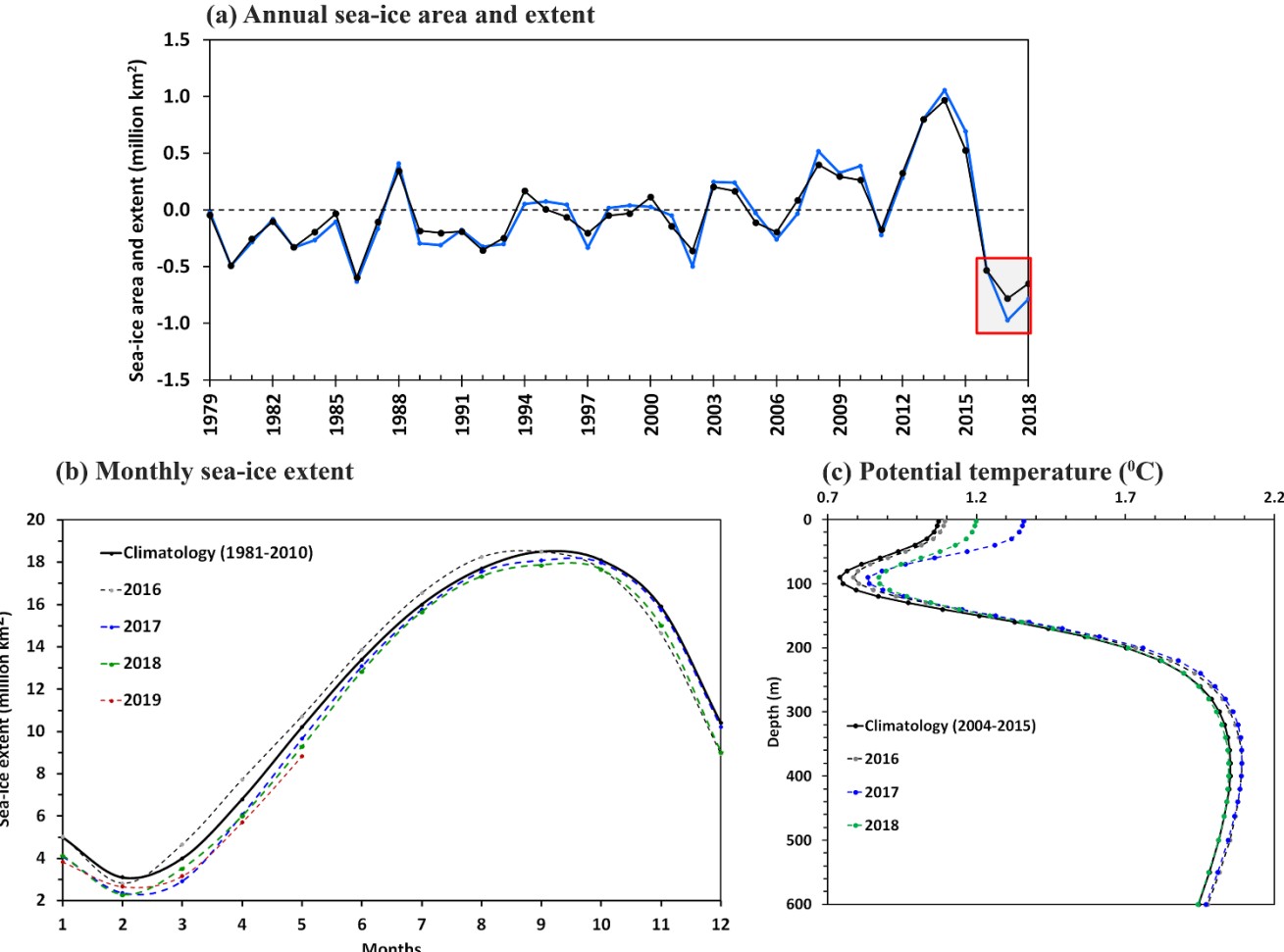

**Figure 9**. (a) Inter-annual variability of Antarctic sea-ice area (black) and extent (blue) anomaly relative to the climatology (1979-2015), as analyzed from satellite observations of passive microwave sensors. Red rectangle shows the anomalous record lowest sea-ice area and extent observed since three successive years from 2016 to 2018 with the maximum melting occurred in 2017. (b) Monthly mean sea-ice extent data indicated loss of sea-ice that started from September 2016 and continued for the year 2017, 2018 and 2019. (c) Argo based ocean potential temperature data (2004-2018) indicated

anomalous upper ocean warming of the Southern Ocean from 2016 to 2018. The potential temperature was spatially averaged over the south of 55°S region encircling the Antarctica.

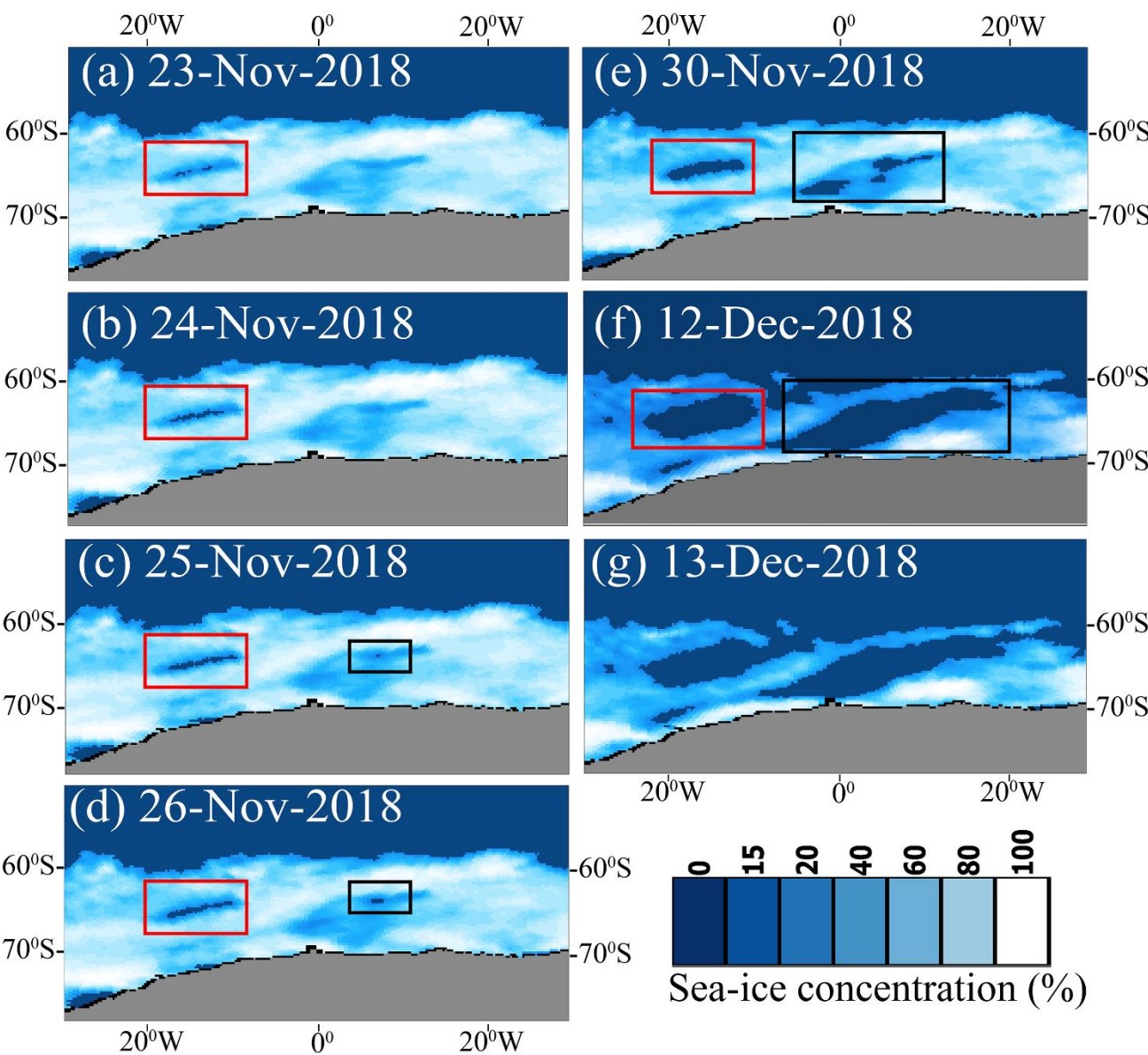

**Figure 10**. The Special Sensor Microwave Imager Sounder (SSMIS) shows the reappearance of the Weddell Sea (red rectangle) and Maud Rise (black rectangle) polynyas starting from 23 November 2018 to 12 December 2018. The polynya disappeared in 13 December 2018.

**Code/Data availability**

We have analyzed monthly sea-ice concentration (SIC) data (September to November 2017) from the passive microwave
sensors with spatial resolution of 25 km acquired from the National Snow and Ice Data Center (NSIDC) (Data id-G02135, Version 3, https://nsidc.org/data). The data were generated using the NASA Team algorithm, which converts satellite derived brightness temperatures to gridded SIC (Cavalieri, D. J., C. L. Parkinson, P. Gloersen, 1997). We used ocean potential temperature data from global marine Argo atlas (http://www.argo.ucsd.edu/Marine_Atlas.html#) that indicated anomalous upper ocean warming of the Southern Ocean from 2016 to 2018. In order to analyze the Aqua-MODIS derived net primary production (NPP), we have validated three ocean-color based models such as the vertically generalized production model (VGPM), *Eppley*-VGPM, and carbon-based productivity model (CbPM) for selecting the best model for the study region. The model based NPP values were available in weekly time scale with a spatial resolution of ~4 km (https://www.science.oregonstate.edu/ocean.productivity). The Argo data are being generated from the Southern Ocean Carbon and Climate Observations and Modeling (SOCCOM) Project funded by the National Science Foundation, Division of Polar Programs (NSF PLR -1425989), supplemented by NASA, and by the International Argo Program and the NOAA programs. The data are available at https://www.mbari.org/science/, http://www.argo.ucsd.edu/, http://argo.jcommops.org/. The primary production data used for the validation are available at https://data.mendeley.com/ data repository under doi: 10.17632/k438knz9zs.5.

**Author contributions**

**BJ.** All works are carried out by BJ except the validation experiment of ocean color data using in-situ observations from the Southern Ocean expeditions.

**NAK.** Validation of ocean color data using in-situ observations from the Southern Ocean expeditions, revision of the manuscript.

**Competing interests**

The authors declare no conflict of interest.

**Acknowledgments**

The authors are thankful to the Director, NCPOR, for his continuous support. The author greatly acknowledges various organizations such as the National Snow and Ice Data Center (NSIDC), National Oceanic and Atmospheric Administration (NOAA), National Aeronautics and Space Administration (NASA) Goddard Space Flight Center (Ocean Biology Processing Group), and their data processing teams for making various datasets available in their portals. Argo data were available from the Southern Ocean Carbon and Climate Observations and Modeling (SOCCOM) Project funded by the National Science Foundation, Division of Polar Programs (NSF PLR -1425989), supplemented by NASA, and by the International Argo

Program and the NOAA programs (http://www.argo.ucsd.edu, http://argo.jcommops.org). We also acknowledge N. Gandhi, IITM, for providing the in-situ primary production data.

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
