# Peer review of "Satellite observations of unprecedented phytoplankton blooms in the Maud Rise Polynya, Southern Ocean"

_The Cryosphere, 2019_

## Short Comment (SC1) · 24 Dec 2019

I would like to give some comments on this article which reports evidence of new phytoplankton blooms and its causative physical mechanism. Firstly, I would suggest to make modification 'unprecedented bloom' instead of new bloom'. The study demonstrates how the phytoplankton over the shallow bathymetric region of the Southern Ocean would likely respond in the future under a warming climate condition and continued melting of Antarctic sea-ice. The authors did a very good work combining variety of remote sensing satellite sensors, Bio-ARGO and reanalysis products. The authors states that the occurrence of phytoplankton bloom over the shallow regions of Maud

[Figure]

Rise seamount where the doming of isotherm/isopycnal brings deeper high nutrient water above the seamount where it may be utilised with a conducive environment of light availability. Why the salinity data is analysed in this case? I suggest the authors to look into the salinity data in the ARGO profiles because the melting of sea-ice leads to the development of shallow mixed layer. It is expected to get low salinity values with increased stability of the water column (generate the stability map). It is required to demonstrate the temperature, salinity, mixed layer profiles, and stability map from ARGO depicting extended features. Otherwise, the reported event is a rare event, and likely to be occurred in the future under a warming environment. The readers will learn something new in this manuscript and has worth to report.
* * *

---

## Referee Comment (RC1) · Anonymous Referee #1 · 27 Jan 2020

In this study, the authors present a phytoplankton bloom first known appearance on the Maud Rise polynya in the Southern Ocean. The authors used various remote sensing images from different broadbands (visible, microwaves,...) as well as modelling and in-situ float data to support their approach which is appreciable. The appearance of the bloom is then discussed according to the bio-physical mechanism at its possible origin as well as in the context of the future under warming conditions. Overall, the work is of good scientific quality and the results are significant.

However, I would have liked the authors to go further into their analysis and discussions of the impact of light availability on the bloom. Indeed, it would have been interesting

to compare the irradiance with known threshold in the literature, for example, in parallel of the SIC evolutions as it can have crucial impacts oh the bloom and growth of phytoplankton.

Minor comments :

line 33. replace a largest by the largest

Figure 1a. - There is too much information in figure 1a resulting as difficulties to see or understand what is displayed.

Figure 1. why is the latitude grid different in figure 1a compared to other Figure 1 panels ?

Figure 1b,c,d,e. the authors chose a criteria of chl-a>=0.8 (line 69). why display chl-a >=1 on the scale instead of 0.8 in these panels ?

Figure 1. Would it be possible to also show the SIC extent ?

Figure 2. same question as Figure 1 on the scale maximum displayed.

Figure 2. Difficulties to distinguish the contours of the bathymetry in 2a and 2b.

---

## Author Comment (AC1) · 1 Feb 2020

We are thankful for your constructive comments and suggested additional analysis that helped to improve the quality of the manuscript.

Reviewer's suggestion 1: I would like to give some comments on this article which reports evidence of new phytoplankton blooms and its causative physical mechanism. Firstly, I would suggest to make modification 'unprecedented bloom' instead of new bloom'.

Authors: Yes, we do agree that 'unprecedented bloom' is appropriate than the 'new

bloom'. We will make change as per your suggestion.

Reviewer's suggestion 2: The study demonstrates how the phytoplankton over the shallow bathymetric region of the Southern Ocean would likely respond in the future under a warming climate condition and continued melting of Antarctic sea-ice. The authors did a very good work combining variety of remote sensing satellite sensors, Bio-ARGO and reanalysis products.

Authors: Thanks for the appreciation. We have used the available dataset to report the unprecedented phytoplankton bloom. And, the study demonstrates how the recent melting of Southern Ocean sea-ice can have impact on the phytoplankton biomass over the shallow bathymetric region.

Reviewer's suggestion 3: The authors states that the occurrence of phytoplankton bloom over the shallow regions of Maud Rise seamount where the doming of isotherm/isopycnal brings deeper high nutrient water above the seamount where it may be utilised with a conducive environment of light availability. Why the salinity data is analysed in this case? I suggest the authors to look into the salinity data in the ARGO profiles because the melting of sea-ice leads to the development of shallow mixed layer. It is expected to get low salinity values with increased stability of the water column (generate the stability map). It is required to demonstrate the temperature, salinity, mixed layer profiles, and stability map from ARGO depicting extended features.

Authors: Thank you so much for your constructive suggestions that helped to improve it further. We have carried out the analysis for temperature, salinity, density, mixed layer and water column stability as per your suggestion (attached: Figure S7). We will include the results in our manuscript after the decision. ARGO float indicated mixed layer warming on the Maud Rise during spring 2016 and 2017 (Figure S7a). The upwelling of high saline and warm water into the mixed layer facilitated the sea-ice melting. The melting of sea-ice leads to the development of shallow mixed layer due to the accumulation of freshwater in the upper ocean. Therefore, we observed lower

values of salinity in the mixed layer with increased stability of the water column (Figure S7b-c) and enhanced phytoplankton biomass.

Reviewer's suggestion 4: Otherwise, the reported event is a rare event, and likely to be occurred in the future under a warming environment. The readers will learn something new in this manuscript and has worth to report.

Authors: It's a rare event and likely to be reoccurred. We strongly agree and believe that it has worth to report.

[Figure]

**(a)**

**(b)**

Potential temperature (°C)

Salinity (psu)

**(c)**

Brunt-Väisälä frequency (s⁻¹)

Figure S7. ARGO float (id-5904468) on the Maud Rise showing (a) potential temperature, (b) salinity, and (c) static stability of the upper ocean. White solid line in each panel shows the variability of mixed layer depth.

**Fig. 1.**

---

## Author Comment (AC2) · 1 Feb 2020

Thank you so much for your constructive comments and suggestions that helped to improve the quality of the manuscript to a great extent.

Reviewer's suggestion 1: In this study, the authors present a phytoplankton bloom first known appearance on the Maud Rise polynya in the Southern Ocean. The authors used various remote sensing images from different broad bands (visible, microwaves,...) as well as modelling and insitu float data to support their approach which is appreciable.

[Figure]

Authors: Thank you so much for the appreciation.

Reviewer's suggestion 2: The appearance of the bloom is then discussed according to the bio-physical mechanism at its possible origin as well as in the context of the future under warming conditions. Overall, the work is of good scientific quality and the results are significant.

Authors: We are very much thankful for the appreciation.

Reviewer's suggestion 3: However, I would have liked the authors to go further into their analysis and discussions of the impact of light availability on the bloom. Indeed, it would have been interesting to compare the irradiance with known threshold in the literature, for example, in parallel of the SIC evolutions as it can have crucial impacts of the bloom and growth of phytoplankton.

Authors: As per your suggestion, we have included analysis of light and diffuse attenuation coefficient (Kd, m-1) for downwelling irradiance at 490nm. The bloom did not appear in September 2017 due to low light condition up to 12.6 Einstein m-2 day-1. The bloom was appeared in October-November 2017 under the influence of improved light condition up to 36.1 and 61.9 Einstein m-2 day-1, respectively for October and November (Table 2). PAR data was analyzed in parallel of the SIC evolutions (Figure S8 and Table 2). However, the sensitivity of light threshold value can be possible only in laboratory controlled condition experiment by considering various cultured groups/species of phytoplankton, which is beyond the scope of the study. We have used science quality remote sensing data, reanalysis, and Argo profiles for reporting the unprecedented phytoplankton bloom to what's possible with such a complex atmosphere-ocean-ice event. High diffuse attenuation coefficient (Kd, m-1) for downwelling irradiance observed up to 0.39 m-1 and 0.37 m-1 during October and November, respectively, which is an indicator of sediment resuspension and bloom condition.

Minor comments :
Reviewer's suggestion 4: line 33. replace a largest by the largest

Authors: Yes, we will replace. Thank you so much.

Reviewer's suggestion 5: Figure 1a. - There is too much information in figure 1a resulting as difficulties to see or understand what is displayed.

Authors: This is due to overlapping of dense ARGO locations and various bathymetric contours. We will surely bring more clarity by excluding few bathymetric contours.

Reviewer's suggestion 6: Figure 1. why is the latitude grid different in figure 1a compared to other Figure 1 panels ?

Authors: We kept the extent purposefully in figure 1a to show the readers about the proximity of Maud Rise to the Antarctica and surrounding seas.

Reviewer's suggestion 7: Figure 1b,c,d,e. the authors chose a criteria of chl-a>=0.8 (line 69). why display chl-a>=1 on the scale instead of 0.8 in these panels ?

Authors: The color bar indicates all values starting from 0.01 to 1. We have labeled 0.8 in the color bar (Figures 1b,c,d,e), so that it will become easy for the readers to follow.

Reviewer's suggestion 8: Figure 1. Would it be possible to also show the SIC extent?

Authors: Since the study region was fully covered by the sea-ice with a polynya on the Maud Rise, the sea-ice concentration (SIC) data found to be appropriate for the analysis. The SIC extent is more useful in the marginal ice zone. However, we have shown the extent of the polynya as dashed polygon in figure 1a. Thank you so much.

Reviewer's suggestion 9: Figure 2. same question as Figure 1 on the scale maximum displayed.

Authors: We will mark at 0.8 mg m-3 in the color bar (Figures 2), so that it will become easy for the readers to follow.

Reviewer's suggestion 10: Figure 2. Difficulties to distinguish the contours of the

bathymetry in 2a and 2b.

Authors: Thank you so much. We will bring better clarity in bathymetric contours of figures 2a and 2b, by changing it to some different color.

[Figure]

[Figure]

Figure S8. Monthly averaged values of photosynthetically available radiation during the appearance of polynya in (a) October and (b) November 2017.

**Table 2**. Net primary production and bio-optical parameters during the occurrence of Maud Rise polynya in October and November 2017. Values for November 2017 are given within brackets. NPP: Net primary production, Chl-*a*: Chlorophyll-*a*, Eu: Euphotic depth, PAR: Photosynthetically available radiation, SST: Sea surface temperature.

| | Minimum | Maximum | Mean | Standard deviation |
|---|---|---|---|---|
| NPP (mg C m$^{-2}$ day$^{-1}$) | 60.08 (101.43) | 374.07 (415.08) | 169.51 (208.44) | 84.04 (50.90) |
| Chl-*a* (mg m$^{-3}$) | 0.07 (0.06) | 3.48 (4.67) | 0.29 (0.28) | 0.26 (0.20) |
| Eu (m) | 27.12 (8.35) | 84.24 (109.56) | 53.72 (56.90) | 13.59 (12.49) |
| PAR (Einstein m$^{-2}$ day$^{-1}$) | 6.27 (13.80) | 36.10 (61.90) | 17.79 (31.43) | 6.86 (8.21) |
| Kd 490 (m$^{-1}$) | 0.03 (0.02) | 0.39 (0.37) | 0.06 (0.06) | 0.03 (0.02) |
| SST (°C) | -1.80 (-1.80) | -1.25 (-1.31) | -1.67 (-1.65) | 0.12 (0.14) |

**Fig. 1.**

---

## Referee Comment (RC2) · Anonymous Referee #2 · 5 Feb 2020

General comments The authors reported phytoplankton blooms in the Maud Rise Polynya, Southern Ocean, which was unseen earlier in spring seasons from entire records of ocean color satellite data acquisition. On the basis of satellite data from CZCS, SeaWiFS, MODIS, and VIIRS they showed the bloom appeared for the first time in the satellite records since 1978. The linkage between the observed bloom and the oceanic $pCO_2$ condition were studied using Argo data. The low $pCO_2$ values in the polynya was possibly due to the presence of chl-a bloom with high NPP, which has potential to drive $CO_2$ fluxes from the atmosphere to the ocean. The observed biological pumping process in the polynya could play an important role for lowering the atmospheric $CO_2$ through transferring of atmospheric $CO_2$ to the ocean. I would

suggest the authors to revise the manuscript according to the following comments.

Specific comments Authors have shown the upwelling of high saline and warm water leads to melting sea-ice. While the conclusions are supported by the evidence, do the early loss of sea ice cover should lead to the warming of mixed layer through radiative heating particularly increase of shortwave radiation in the ocean surface in spring period? So the authors are suggested to look into any unusual enhancement of shortwave radiation in the polynya region during the study period compared to the long-term prevailed condition?? Any satellite or reanalysis/modeling product is adequate for this analysis.

Even though some of the works are now included by the authors for the analysis of mixed layer warming (comments from other reviewers) on the Maud Rise, the short-wave heat input in absence of ice cover is crucial. Yet the manuscript is structured superbly with scientific understanding and hard to find technical flaws. Some data are known to have uncertainty in the polar waters specifically remote sensing based primary productivity (level-4) viz. vgpm, eppey-vgpm, cbpm models, that lacking validation with ship measurements. But the uncertainties in remote sensing methods are apparently quantified by the authors using in-situ NPP estimated using 13C tracer from the Indian scientific expedition to the polar waters.

Other minor comments: Line 15: Make expansion of 'MODIS' Line 140: It should be 'covered by the sea-ice' Line 170: Describe about the white contours in figure 2. Line 230: Are there any studies that quantifies carbon fixation by prymnesiophytes (Phaeocystis antarctica) and diatoms in Antarctica sea ice? Line 230: Diffuse attenuation coefficient at 490 may be included here (see the comments from the other author) and discuss in the text.

---

## Author Comment (AC3) · 9 Feb 2020

Reviewer's suggestion 1: General comments: The authors reported phytoplankton blooms in the Maud Rise Polynya, Southern Ocean, which was unseen earlier in spring seasons from entire records of ocean color satellite data acquisition. On the basis of satellite data from CZCS, SeaWiFS, MODIS, and VIIRS they showed the bloom appeared for the first time in the satellite records since 1978. The linkage between the observed bloom and the oceanic pCO2 condition were studied using Argo data. The low pCO2 values in the polynya was possibly due to the presence of chl-a bloom with high NPP, which has potential to drive CO2 fluxes from the atmosphere to the

ocean. The observed biological pumping process in the polynya could play an important role for lowering the atmospheric CO2 through transferring of atmospheric CO2 to the ocean. I would suggest the authors to revise the manuscript according to the following comments.

Authors: Thanks for your insightful comments and we will revise the manuscript accordingly.

Reviewer's suggestion 2: Specific comments: Authors have shown the upwelling of high saline and warm water leads to melting sea-ice. While the conclusions are supported by the evidence, do the early loss of sea ice cover should lead to the warming of mixed layer through radiative heating particularly increase of shortwave radiation in the ocean surface in spring period? So the authors are suggested to look into any unusual enhancement of shortwave radiation in the polynya region during the study period compared to the long-term prevailed condition?? Any satellite or reanalysis/modeling product is adequate for this analysis.

Authors: As per the suggestion, we have analyzed the shortwave radiation for looking into any unusual enhancement of values in the polynya region. We computed the anomaly of shortwave radiation in September-November 2017 from the long-term mean of 1979-2015. Analysis indicated record highest gain of shortwave radiation in the polynya region during September-November 2017, considering the 38-year time series starting from 1979 through 2016 (Figure. S9). The observed anomalous gain is possibly due to the early loss of sea ice cover. The results will be included in the revised version. Thank you so much for your suggestion that has helped to improve the quality of the manuscript

Reviewer's suggestion 3: Even though some of the works are now included by the authors for the analysis of mixed layer warming (comments from other reviewers) on the Maud Rise, the shortwave heat input in absence of ice cover is crucial. Yet the manuscript is structured superbly with scientific understanding and hard to find technical flaws. Some data are known to have uncertainty in the polar waters specifically remote sensing based primary productivity (level-4) viz. vgpm, eppey-vgpm, cbpm models, that lacking validation with ship measurements. But the uncertainties in remote sensing methods are apparently quantified by the authors using in-situ NPP estimated using 13C tracer from the Indian scientific expedition to the polar waters.

Authors: We do agree that the shortwave heat input in the polynya is crucial, and therefore the analysis has been carried out as per your suggestion (Figure. S9). In fact, in-situ net primary production (NPP) observations are rarely available for the Southern Ocean. The main source of NPP data for the Southern Ocean is the ocean-colour-based models widely used in the scientific community. We evaluated the performance of these models by comparing with the in-situ NPP estimated using 13C tracer during the Indian scientific expeditions to the Southern Ocean.

Other minor comments: Reviewer's suggestion 4: Line 15: Make expansion of 'MODIS'

Authors: We will expand as Aqua-Moderate Resolution Imaging Spectroradiometer.

Reviewer's suggestion 5: Line 140: It should be 'covered by the sea-ice'

Authors: We will correct it.

Reviewer's suggestion 6: Line 170: Describe about the white contours in figure 2.

Authors: We will describe about the white contours as bathymetry in figures 2.

Reviewer's suggestion 7: Line 230: Are there any studies that quantifies carbon fixation by prymnesiophytes (Phaeocystis antarctica) and diatoms in Antarctica sea ice?

Authors: As per our knowledge, there is no comprehensive study that quantifies carbon fixation by prymnesiophytes (Phaeocystis antarctica) and diatoms in the Antarctica sea ice.

Reviewer's suggestion 8: Line 230: Diffuse attenuation coefficient at 490 may be included here (see the comments from the other author) and discuss in the text.

Authors: We will discuss about the diffuse attenuation coefficient in the appropriate place.

[Figure]

Figure S9. Monthly anomalies of shortwave radiation for (a) September, (b) October, and (c) November 2017 in the Maud Rise polynya (black rectangles). The anomalies were computed relative to a 38-year climatology (1979-2016). The regions within grey polylines shows the record level shortwave radiation in 2017 that lies outside of shortwave radiation values from 1979 to 2016.

**Fig. 1.**

---

## Author Response (AR1)

**Comments to the Author:**

Comment 1. Based on the reviews, I recommend this paper be accepted for publication in TC after minor revisions. In addition to addressing the reviewer comments, I have a few suggestions to improve the readability of the paper:

**Authors.** Thank you so much for your constructive comments. We have revised the manuscript accordingly with reference to your comments, two anonymous reviewers, and public review.

Comment 2. I strongly discourage the extensive use of supplemental material in this case. Supplemental material is best used for material that cannot be put in the main paper (e.g. multimedia) or technical details. As your supplemental figures are relevant to the main results and discussion, the readability of the paper would be improved if this material was included in the main paper instead. This will make your paper more accessible to readers.

**Authors.** As per the suggestion, one of the important figure (Supplementary Fig. S4) much relevant to the main result is now moved into the main text as Fig. 5 of the revised manuscript.

Comment 3. Line 85-95 and table 1: What in-situ data are you validating against? This needs to be described and referenced.

**Authors.** This is included in the revised manuscript. The pixel values from the models were extracted around each in-situ observations of NPP to generate the matchups for the validation strategy. The details of in-situ measurement ($^{13}C$ method) is documented in the previous work (Gandhi et al., 2012).

Comment 4. Materials and Methods – here you list what data you use, but it's not clear in what way it is used. E.g. the Argo float data, and eddy data. This section could be improved if you simply state how each of these data sets will be used.

**Authors.** This section is improved in the revised manuscript by including how the datasets are being used.

Comment 5. Line 142 – "missing" might imply you had no data, but I think you mean that no chl-a was observed.

**Authors.** Modified the sentence as 'no chl-a wa observed' in place of 'missing'.

Comment 6. Line 157 – If it's only one float, it's not a fleet.

**Authors.** Removed the word 'fleet of' in the revised manuscript.

Comment 7. Figure S9 – please be clear if this is incident shortwave or absorbed shortwave. Also, please indicate where this data is from in the revised manuscript.

**Authors.** Yes, this is an incident shortwave radiation which has been specified in the revised manuscript. The revised figure number is now Figure-S6. The monthly incident shortwave radiation was acquired from the European Center for Medium-Range Weather Forecast (ECMWF) at grid resolution of 0.25° during January 1979-December 2017. Monthly anomalies of shortwave radiation for September-November 2017 was computed relative to a 38-year climatology (1979-2016). The details are incorporated in the revised text.

Comment 8. Figure S6 – specify which is Weddell Sea polynya and which is Maud Rise Polynya.

**Authors.** The figure is revised as per the suggestion. Reappearance of the Weddell Sea polynya is shown within the red rectangles and Maud Rise as black rectangles. The revised figure number is now Figure-S8.

Comment 9. Non-public comments to the Author:

There is numerous instances of awkward English phrasing and syntax throughout the manuscript that impacts the readability. If accepted, the manuscript will be copyedited before publication, but I recommend a careful reading through and editing to correct as many of these as possible now to ensure your intended meaning is clear.

**Authors.** We have tried to improve for the English phrasing so as to convey the intended meaning in the manuscript. Thank you so much for your suggestion that has indeed helped to improve the quality of the manuscript.

[revised manuscript text omitted]
 S45.** Hydrographic profiles (potential temperature, salinity, density, mixed layer and water column stability) from an Argo float (ID-5904468) on the Maud Rise starting from 2015 through 2019. The profiles indicated mixed layer warming on during spring 2016 and 2017. The upwelling of high saline and warm water into the mixed layer facilitated the sea-ice melting. The melting of sea-ice leads to the development of shallow mixed layer due to the accumulation of freshwater in the upper ocean. Therefore, we observed lower values of salinity in the mixed layer with increased stability of the water column. The development of shallow mixed layer improved the light availability in the upper ocean and the condition is favourable for the growth of phytoplankton. The Argo data are being generated from the Southern Ocean Carbon and Climate Observations and Modeling (SOCCOM) Project funded by the National Science Foundation, Division of Polar Programs (NSF PLR -1425989), supplemented by NASA, and by the International Argo Program and the NOAA programs. The data are available at https://www.mbari.org/science/.

**Text S5.** Monthly averaged values of photosynthetically available radiation were acquired from Aqua-MODIS during the appearance of polynya in October and November 2017. The bloom was appeared in October-November 2017 under the influence of improved light condition up to 36.1 and 61.9 Einstein m-2 day-1, respectively for October and November.

[revised manuscript text omitted]

**PAR (Einstein m-2 day-1)**

**Fig. S5**6 Monthly averaged values of Aqua-MODIS photosynthetically available radiation during the appearance of polynya in (a) October and (b) November 2017.

[Figure]

125

Fig. S6 Monthly anomalies of incident shortwave radiation for (a) September, (b) October, and (c) November 2017 in the Maud Rise polynya (black rectangles). The anomalies were computed relative to a 38-year climatology (1979-2016). The regions within grey polylines shows the record level shortwave radiation in 2017 that lies outside of shortwave radiation values from 1979 to 2016. The data were acquired from the European Center for Medium-Range Weather Forecast (ECMWF) at a grid resolution of 0.25°.

130

[Figure]

135

**Fig. S7** Argo data were utilized to find the relationship between the chlorophyll-*a* and the oceanic $pCO_2$ condition. (a) The coefficient of correlation (*r*) between the $pCO_2$ and chl-*a* was found to be -0.56 ($p < 0.01$) during August-September 2017. (b) The relationship improved ($r = -0.82$, $p < 0.01$) during the bloom condition in October-November 2017. The best relationship observed between the $pCO_2$ and chl-*a* when the data was log transformed ($r = -0.94$, $p < 0.01$).

140

[Figure]

[Figure]

**Fig. S86** The Special Sensor Microwave Imager Sounder (SSMIS) shows the reappearance of the Weddell Sea (red rectangle) and Maud Rise polynya (within theblack rectangle) polynyas starting from 23 November 2018 to 12 December 2018. The polynya disappeared in 13 December 2018.

---

## Author Response (AR2)

**Comments to the Author:**

1. The authors have addressed the reviewer comments, but have not fully addressed the editor's recommendations on presentation. Therefore, I will make detailed recommendations for changes in presentation to make the manuscript consistent with TC style and standards.

Supplemental material should only be used in very specific circumstances. It cannot be used to illustrate new points. It is intended for items that cannot fit in the main text because of format or size, but should be avoided in other cases. In most cases, the SM figures illustrate new points. In these cases, they MUST be in the main text. In those few cases where the supplement points are already made in the main paper with text and figures (see below with respect to Figure 5, S3 and S4), I still recommend placing in the main text, as there is no reason for a simple figure to be in SM. In this case, an appendix might be reasonable, though I do not see that as necessary here.

These changes will make the paper more readable. The authors should either put all SM figures in the main text, or if not needed to support any points, leave them out altogether.

**Authors.** Thank you so much for your suggestions. As per the suggestion, we have placed all the figures in the main text to make the manuscript consistent with the journal policy. Now, there is no supplementary figures and texts in the revised manuscript.

2. Figure S1 – Figures should not normally be in the introduction. This presents results similar to what has been previously published (at least with respect to sea ice, to some degree with the ARGO data). If these results are adequately published elsewhere, I recommend it just be cited with no figure if you are citing it in the Introduction. If these are new results, these SHOULD NOT be in the supplemental material, nor in the introduction and should be presented in the Results section with a figure in the main text. This was not commented on by the referees, so I will leave this up to the authors, but I suggest this would be more useful if you just looked at the Weddell Sea rather than the whole Antarctic, and then this could be included in the results.

**Authors.** As per the suggestion, we have removed the Figure S1 in the introduction section and cited Parkinson (2019) for discussing recent sea-ice changes. Figures S1 is moved in the main text (Figure 9) to show the likelihood of Antarctica polynya occurrence with a background of anomalous upper ocean warming and sea-ice loss, similar to the events that occurred in the Antarctic sea-ice from 2016 to 2019.

3. Figure S2 – These data are useful to understand how close the data are to the study area, and the quality of the relationship with NPP. As this is used to choose the model, it should be in the main text.

**Authors.** Included in the main text as Figure 3.

4. Figure S3, S4, and Figure 5: These figures all support the same point. Notably, S4 and 5 show the same data, with Fig 5 just showing detail. I'd combine into a single figure, but it is fine if you want to keep them separate. But as they are presenting evidence for the new point that there is upwelling locally at the MR, they should both be in the main text. Figure S3 isn't consistent in timing with Figure 5, and since the former is a model and latter data, I wonder if S3 is needed to

make your point and should just be excluded. If you retain it, I think it should be discussed a bit more in the main text to explain these differences (and thus, the figure needs to be in the main text).

**Authors.** As per the suggestion, we have removed Figure S3. And, we merged Figure S4 and Figure 5 which are included in the main text as Figure 6.

5. Figure S6 – again, this is introducing a new point, thus according to journal policy, should be in the main text.

**Authors.** Included in the main text as Figure 7.

6. Figure S7 – same reasoning as for S6.

**Authors.** Included in the main text as Figure 8.

7. Figure S8 – This figure is pertinent to show the timing of the polynya, so should be in the main text. I'd suggest that since this provides a clear visual of the evolution of the physical setting driving your observations, that this would be most useful as a figure 1 to provide the reader an introduction to the setting.

**Authors.** Now, this figure is included in the main text to show the reappearance of polynya in 2018. The evolution of polynya in 2017 and its physical setting is shown in Figures 3a-c and 1a.

8. Non-public comments to the Author:
I have sought clarification on journal style with respect to supplemental material. In this case, your SM figures do not meet the standard for use in this case. Therefore, I am making the decision that all figures and text should be in the main paper before it can be published.

There remain widespread non-standard phrasing throughout the paper. I would again urge the authors to take a careful look through and edit to avoid misunderstandings and delays when the paper goes through copy-editing. I have not taken the time myself to provide suggestions in order to expedite the process.

**Authors.** Sir, thank you so much for your suggestions. We have moved all the supplementary figures and texts into the main text as per the journals policy. We have tried to improve for the English phrasing so as to convey the intended meaning.

---

## Author Response (AR3)

**Comments to the Author:**

**Editor:** One final point that should be clarified - The Figure 7 caption indicates the figure is showing incident shortwave radiation, which one would normally interpret as incident downwelling shortwave. The text on line 233 indicate net shortwave (downwelling-upwelling), which would be the absorbed shortwave. I think the figure is showing net. If so, the caption should be fixed

**Authors:** Yes, the Figure 7 indicates the net shortwave that we have corrected in the caption. Thank you so much for your suggestion.